# A Complex Structure for Two-Typed Tangent Spaces

**DOI:** 10.3390/e27020117

**Published:** 2025-01-24

**Authors:** Jan Naudts

**Affiliations:** Physics Department, Universiteit Antwerpen, Universiteitsplein 1, 2610 Antwerpen, Belgium; jan.naudts@uantwerpen.be

**Keywords:** complexified tangent spaces, parallel transport, modular operator, complex connection coefficients, Kubo–Mori theory, admittance function, fluctuation–dissipation theorem

## Abstract

This study concerns Riemannian manifolds with two types of tangent vectors. Let it be given that there are two subspaces of a tangent space with the property that each tangent vector has a unique decomposition as the sum of a vector in one subspace and a vector in the other subspace. Then, these tangent spaces can be complexified in such a way that the theory of the modular operator applies and that the complexified subspaces are invariant for the modular automorphism group. Notions coming from Kubo–Mori theory are introduced. In particular, the admittance function and the inner product of the Kubo–Mori theory can be generalized to the present context. The parallel transport operators are complexified as well. Suitable basis vectors are introduced. The real and imaginary contributions to the connection coefficients are identified. A version of the fluctuation–dissipation theorem links the admittance function to the path dependence of the eigenvalues and eigenvectors of the Hamiltonian generator of the modular automorphism group.

## 1. Introduction

Almost half a century ago, Rieffel and van Daele [1] gave a geometric interpretation of the theory of the modular operator, also known as Tomita–Takesaki theory [2]. One of the origins of this theory is the relation between left and right Haar measures, as treated, for instance, in the book by Halmos [3]. An analog is the symmetry between the left and right multiplication of non-commuting operators. In the geometric approach, this symmetry is realized by taking the orthogonal complement of the subspaces of a real Hilbert space.

In a recent paper [4], the geometric approach was reconsidered with an emphasis on the complex structure that is determined by the two subspaces of a real Hilbert space, even in the case that these subspaces have unequal dimensions.

In the present work, this theory is used to embed the tangent spaces of a Riemannian manifold in complex Hilbert spaces. It is assumed that the tangent spaces of the manifold can be decomposed into two subspaces. The elements of the subspaces are said to be typed. The theory of the modular operator can then be applied [1,4] to complexify the tangent spaces in such a way that the modular automorphism group leaves the subspaces invariant.

A simple example of a manifold with two-typed tangent spaces is spacetime in relativistic mechanics. In spacetime, there are three directions of spacelike vectors and one timelike direction. In the literature, a complexified spacetime is usually found in combination with a Wick rotation. Note that the Kubo–Martin–Schwinger (KMS) condition [5] discussed below in Section 6 is an implementation of such a Wick rotation. The parameter τ along the real axis of a complex plane is linked by analytic continuation to a parameter, β, along the imaginary axis. In Statistical Physics, the parameter τ represents the time, while β is an inverse temperature.

Another example of 3 + 1 typing is found in Section 3.10 of [4]. In that example, the complexification of the tangent spaces reproduces the quantum mechanics of Larmor precession.

The formulation of the KMS condition and of the modular operator theory in the second half of the twentieth century were preceded by the fundamental breakthrough of Kubo’s insight [6] that non-equilibrium phenomena in Statistical Physics, which are usually accompanied by the dissipation of energy, can be linked to spontaneous fluctuations occurring in equilibrium states. This led to what is known as Kubo–Mori theory [7,8]. The present work shows that this link between non-equilibrium phenomena and equilibrium fluctuations can be understood in the context of Differential Geometry. Fluctuations as a function of time induce time-dependent correlations which are captured by the linear response function $\phi_{xy}\left(\tau \right)$ and its transform, the admittance function $\chi_{xy}\left(w\right)$. On the other hand, the time evolution of a system not in thermodynamic equilibrium results in changes in the eigenvalues and eigenvectors of the Hamiltonian H(θ), the generator of the modular automorphism group, as θ moves along a path, γ, in the manifold M of thermodynamic states.

The main result of the present work is the relation (Equation 25) between the evolution of the eigenvalues and eigenvectors of the modular operator $\hat{\Delta }$ along the path γ and an expression containing the admittance function. Methods to calculate evolving eigenvalues and eigenvectors in the context of Solid-State Physics are reviewed in [9]. In the present work, the context is more abstract and the emphasis is on the underlying geometry.

The structure of this paper is as follows. Section 2, Section 3, Section 4 and Section 5 recall known facts about the two subspaces of a real Hilbert space and the complex structure resulting from them. Section 6, Section 7 and Section 8 discuss the KMS condition and Linear Response Theory. The admittance function is introduced. Perturbations of the Hamiltonian are considered.

Starting from Section 9, the parameter dependence of the tangent spaces is studied. The parallel transport operators are complexified. A suitable basis of tangent vectors is introduced in Section 10. In this basis, the connection coefficients are obtained by considering infinitesimal parallel transport. Section 12 presents a fluctuation–dissipation theorem. The final section contains a short discussion of the results obtained so far.

## 2. Two-Typed Spaces

Let it be given that there are two closed subspaces $K_{\theta }$ and $L_{\theta }$ of the tangent space $T_{\theta }M$ of a finite-dimensional parameterized Riemannian manifold, M. Assume that they have the trivial intersection $K_{\theta }\cap L_{\theta }=\left\{0\right\}$ and that the sum $K_{\theta }+L_{\theta }$ of the two equals $T_{\theta }M$. It is assumed throughout the text that the tangent space $T_{\theta }M$ is finite-dimensional. If this is not the case, then the appropriate assumption is that $K_{\theta }+L_{\theta }$ is dense in $T_{\theta }M$. Any tangent vector, x, in $T_{\theta }M$ has a unique decomposition:$$x=u+v\;\mathrm{with}\;u\in K_{\theta }\;\mathrm{and}\;v\in L_{\theta }.$$

For the simplicity of the notations, the index θ is dropped further on when the focus is on a single tangent space.

An important result of [4], originally found in [1], is the existence of a unique orthogonal operator, *J*, which is Hermitian and has properties such that$$JK=L^{⊥}\;\mathrm{and}\;JL=K^{⊥}$$
and that for any u∈K and v∈L, one has(Ju,u)≥0,(Jv,v)≤0and(Ju,v)=0.

Here, (x,y) is the real inner product of the vectors x and y in the tangent space. $K^{⊥}$ and $L^{⊥}$ are the orthogonal complements of K and L, respectively.

Note that a pseudometric *G* can be defined byG(x,y)=(Jx,y).

It takes positive values for x=y in K and negative values for x=y in L and vanishes if x and y are of pure but opposite types.

## 3. The Modular Operator

A linear operator, *S*, is defined in the real Hilbert space by(1)S(u+v)=u−vu∈K,v∈L.

It defines a positive operator, Δ, called the *modular operator,* through $\Delta =S^{†}\;S$. Here, $S^{†}$ is the Hermitian conjugate of *S* in the real Hilbert space, $T_{\theta }M$. The isometry *J*, introduced earlier, shows up in the polar decomposition of the operator *S*. Indeed, one can prove [1,4] that(2)$$S=J\;\Delta^{1/2}$$
and hence that $\Delta^{1/2}=JS$.

## 4. Complexified Tangent Spaces

In [4], the complex number *i* is defined in $T_{\theta }M\times T_{\theta }M$ by(3)$$i=\left(\begin{array}{cc} 0 & J\\ -J & 0 \end{array}\right).$$
with *J* being the operator introduced in the previous sections. It turns the product space $T_{\theta }M\times T_{\theta }M$ into a complex Hilbert space, $H_{\theta }$. The inner product ${\left(x,y\right)}_{\theta }$ of two elements, x,y of $H_{\theta }$, is determined by the requirement that it coincides with the real-valued inner product when x and y in $T_{\theta }M$ are identified with ${\left(x,0\right)}^{T}$ and ${\left(y,0\right)}^{T}$, respectively, in $H_{\theta }$.

Let$$K^{⊕}=K\times K^{⊥}\subset T_{\theta }M\times T_{\theta }M.$$

It is the real subspace of H, spanned by elements of the form ${\left(u,v\right)}^{T}$ with u in K and v in $K^{⊥}$. Similarly, let$$L^{⊕}=L\times L^{⊥}\subset T_{\theta }M\times T_{\theta }M.$$

One has$$iK^{⊕}=L^{⊕}.$$

The complex Hilbert space H is spanned by these two real subspaces, i.e.,$$H=K^{⊕}+L^{⊕}.$$

This means that any vector, x, of H can be written as x=u+iv with u and v in $K^{⊕}$. In addition, one can show [4] that the intersection of any two of the spaces, $K^{⊕}$, $L^{⊕}$, ${\left(K^{⊕}\right)}^{⊥}$ and ${\left(L^{⊕}\right)}^{⊥}$, is trivial.

The operator $\hat{S}$ is now defined by(4)$$\hat{S}\;\left(u+iv\right)=u-iv,\;u,v\in K^{⊕}.$$

Its polar decomposition is written as$$\hat{S}=\hat{m}\;{\hat{\Delta }}^{1/2}.$$

One can show [4] that the anti-linear isometry $\hat{m}$ and the modular operator $\hat{\Delta }$ are given, respectively, by(5)$$\hat{m}=\left(\begin{array}{cc} J & 0\\ 0 & -J \end{array}\right)\;\mathrm{and}\;\hat{\Delta }=\left(\begin{array}{cc} \Delta & 0\\ 0 & \Delta^{-1} \end{array}\right).$$

## 5. Eigenvalues

The eigenvectors of the modular operator $\hat{\Delta }$ are needed in what follows. They are characterized by the following proposition.

**Proposition** **1.**
*A vector, x, of H is an eigenvector of the modular operator $\hat{\Delta }$ with the eigenvalue λ if and only if $\left(1+\sqrt{\lambda }\;\hat{m}\right)\;x$ belongs to $K^{⊕}$ and $\left(1-\sqrt{\lambda }\;\hat{m}\right)\;x$ belongs to $L^{⊕}$.*


**Proof.** If x is an eigenvector of $\hat{\Delta }$ with the eigenvalue λ, then one has$$\begin{array}{cc} \left(1+\sqrt{\lambda }\;\hat{m}\right)\;x & =\left(1+\hat{m}\Delta^{1/2}\right)\;x=\left(1+\hat{S}\right)\;x. \end{array}$$This vector belongs to $K^{⊕}$ because $\left(1+\hat{S}\right)/2$ projects onto $K^{⊕}$.Similarly, $\left(1-\sqrt{\lambda }\;\hat{m}\right)\;x$ belongs to $L^{⊕}$ because $\left(1-\hat{S}\right)/2$ projects onto $L^{⊕}$.Conversely, if $\left(1+\sqrt{\lambda }\;\hat{m}\right)\;x$ belongs to $K^{⊕}$ and $\left(1-\sqrt{\lambda }\;\hat{m}\right)\;x$ belongs to $L^{⊕}$, then one has$$\begin{array}{cc} {\hat{\Delta }}^{1/2}\;x & =\hat{m}\hat{S}\;x\\ \; & =\frac{1}{2}\hat{m}\hat{S}\left(1+\sqrt{\lambda }\;\hat{m}\right)\;x+\frac{1}{2}\hat{m}\hat{S}\left(1-\sqrt{\lambda }\;\hat{m}\right)\;x\\ \; & =\frac{1}{2}\hat{m}\left(1+\sqrt{\lambda }\;\hat{m}\right)\;x-\frac{1}{2}\hat{m}\left(1-\sqrt{\lambda }\;\hat{m}\right)\;x\\ \; & =\sqrt{\lambda }\;x. \end{array}$$This shows that x is an eigenvector with the eigenvalue λ. □

## 6. The KMS Condition

The KMS condition [5] is usually formulated in the context of $C^{*}$ algebras and von Neumann algebras. See, for instance [10]. The definition given in [1] does not refer to the algebras of operators in a complex Hilbert space; it refers only to the real subspaces of the Hilbert space.

**Definition** **1.**
*A strongly continuous one-parameter unitary group, $\left\{U_{\tau }:\tau \in R\right\}$, in the complex Hilbert space H is said to satisfy the KMS condition with respect to the real subspace K of H if for any pair of elements, x,y, of K, there exists a complex function, $F^{xy}\left(w\right)$, defined, bounded and continuous on the strip −1≤Imw≤0 and analytic on the interior, with boundary values given by*

$$F^{xy}\left(\tau \right)=\left(U_{\tau }x,y\right)\;and\;F^{xy}\left(\tau -i\right)=\left(y,U_{\tau }x\right)$$

*for all real values of τ.*


The main result of [1] implies the following.

**Theorem** **1.**
*Let $K^{⊕}$ and $\hat{\Delta }$ be as in the previous sections. The group $\left\{{\hat{\Delta }}^{i\tau }:\;\tau \in R\right\}$ is the unique, strongly continuous one-parameter group of unitaries in the complex Hilbert space H that carries the real subspace $K^{⊕}$ onto itself and satisfies the KMS condition with respect to $K^{⊕}$.*


This group is called the *modular automorphism group*.

## 7. Linear Response and Admittance

For x,y in $K^{⊕}$, the *linear response function* $\phi_{xy}\left(\tau \right)$ is defined by the following. (In [8], the inner product is linear in the second argument. This explains small differences between the expressions here and in [8]).(6)ϕxy(τ)=2iIm(Δ^iτx,y),τ∈R.

Introduce the *admittance function* $\chi_{xy}\left(w\right)$, defined by(7)χxy(w)=i∫0+∞eiwτdτϕxy(τ),Imw>0.

It is the Laplace transform of the linear response function rotated by 90 degrees in the complex plane. It is a retarded Green function [11]. The quantity is important in Physics because it is often accessible for experimental evaluation.

**Proposition** **2.**
*The admittance $\chi_{xy}\left(w\right)$ satisfies*

(8)
χxy(w)=ewΔ^−1w+logΔ^x,y,x,y∈K⊕, Imw>0.



**Proof.** For x,y in $K^{⊕}$, Theorem 1 states that there exists a complex function, $F^{xy}\left(w\right)$, bounded and continuous on the strip −1≤Imw≤0 and analytic on the interior, such that$$\chi_{xy}\left(w\right)=i\int_{0}^{+\infty }e^{iw\tau }\;\;d\tau \;\left[F^{xy}\left(\tau \right)-F^{xy}\left(\tau -i\right)\right].$$Through complex integration around a closed loop in the complex plane, one then obtains$$\begin{array}{cc} \chi_{xy}\left(w\right)\; & =-i\int_{-i}^{0}\;d\tau \;e^{iw\tau }\;F^{xy}\left(\tau \right)\\ \; & =\int_{0}^{1}\;d\beta \;e^{\beta w}\;F^{xy}\left(-i\beta \right)\\ \; & =\int_{0}^{1}\;d\beta \;e^{\beta w}\;\left({\hat{\Delta }}^{\beta }\;x,y\right). \end{array}$$This implies (Equation 8). □

As a consequence of the above result, one can define an inner product, ${\left(x,y\right)}^{∼}$, using(9)$${\left(x,y\right)}^{∼}=\chi_{xy}\left(0\right)=\left(\hat{T}\;x,\hat{T}\;y\right),\;x,y\in K^{⊕}.$$
with $\hat{T}$ being the positive square root of(10)$${\left(\hat{T}\right)}^{2}=\frac{\hat{\Delta }-1}{\log\hat{\Delta }}=\int_{0}^{1}\;d\beta \;{\hat{\Delta }}^{\beta }.$$

The inner product extends through complex linearity/conjugate linearity to all of H.

This inner product is used in the Kubo–Mori theory of linear response [8]. Its importance in the context of the manifolds of density matrices follows from being the unique metric [12,13,14] with the property of monotonicity with respect to completely positive trace-preserving maps and with the property that the e- and m-connections [15] are each other dual with respect to this metric.

## 8. Perturbations

A self-adjoint Hamiltonian, $\hat{H}$, is defined by $\hat{H}=\log\hat{\Delta }$. Add to this Hamiltonian a Hermitian operator, $\hat{B}$, multiplied with a small real number, ϵ. Then, the perturbed modular operator equals(11)$${\hat{\Delta }}^{pert}=\exp(\hat{H}+\epsilon \hat{B}).$$

One has$$\frac{\;d\;}{\;d\epsilon }{\hat{\Delta }}^{pert}{|}_{\epsilon =0}=\int_{0}^{1}\;dr\;{\hat{\Delta }}^{r}\;\hat{B}\;{\hat{\Delta }}^{1-r}.$$

A proof of the identity which was used to derive this expression is found in [15], p. 156.

In the $C^{*}$ algebraic context of [8], the vector Ω is an eigenvector of the modular operator, and the corresponding eigenvalue equals 1. It is not clear whether in the present context 1 is always an eigenvalue. Let us therefore continue by selecting an eigenvector, x, of $\hat{\Delta }$ with the eigenvalue λ. Decompose $\hat{B}x=u+iv$ with u and v in $K^{⊕}$. Then, one has for any y in $K^{⊕}$(12)$$\begin{array}{cc} \left(\frac{\;d\;}{\;d\epsilon }{\hat{\Delta }}^{pert}{|}_{\epsilon =0}\;x,y\right) & =\int_{0}^{1}\;d\beta \;\lambda^{1-\beta }\left({\hat{\Delta }}^{\beta }\;\hat{B}x,y\right)\\ \; & =\lambda \chi_{uy}\left(-\log\lambda \right)+i\lambda \chi_{vy}\left(-\log\lambda \right). \end{array}$$

If the eigenvalue λ equals 1, which is the case in Statistical Physics, then the above result becomes(13)$$\begin{array}{cc} \left(\frac{\;d\;}{\;d\epsilon }{\hat{\Delta }}^{pert}{|}_{\epsilon =0}\;x,y\right) & =\chi_{uy}\left(0\right)+i\chi_{vy}\left(0\right)\\ \; & ={\left(\hat{B}x,y\right)}^{∼}. \end{array}$$

This relation expresses the effect on the modular operator $\hat{\Delta }$ of a perturbation, $\hat{B}$, of the Hamiltonian $\hat{H}$ in terms of the inner product (Equation 9).

## 9. Parallel Transport Operators

From here on, the parameter dependence of the tangent spaces is made explicit again.

A connection, Π, in the manifold M can be defined [16] by a collection of parallel transport operators, $\Pi {\left(\gamma \right)}_{s}^{t}$. They transport vectors from the tangent space $T_{s}M$ to the tangent space $T_{t}M$ along the smooth curve $\gamma :t\mapsto \gamma_{t}$ in the manifold M. Note that $T_{t}M$ stands for $T_{\gamma_{t}}M$; whenever $\gamma_{s}$ or $\gamma_{t}$ appears as an upper or lower index, it is replaced by *s* or *t*, respectively. The obvious requirements are that $\Pi {\left(\gamma \right)}_{t}^{s}$ is the inverse of $\Pi {\left(\gamma \right)}_{s}^{t}$ and that the composition law$$\Pi {\left(\gamma \right)}_{s}^{t}\;\Pi {\left(\gamma \right)}_{r}^{s}=\Pi {\left(\gamma \right)}_{r}^{t}$$
holds along any non-self-intersecting path, γ. In addition, the derivatives of $\Pi {\left(\gamma \right)}_{s}^{t}$ along *t* should exist in some sense. Covariant derivatives are defined by considering infinitesimal parallel transport. They are treated in Section 11.

Complex linear parallel transport operators are defined by$$\hat{\Pi }{\left(\gamma \right)}_{s}^{t}=\left(\begin{array}{cc} \Pi {\left(\gamma \right)}_{s}^{t} & 0\\ 0 & J_{t}\;\Pi {\left(\gamma \right)}_{s}^{t}\;J_{s} \end{array}\right).$$

The inverse of $\hat{\Pi }{\left(\gamma \right)}_{s}^{t}$ is $\hat{\Pi }{\left(\gamma \right)}_{t}^{s}$, and $\hat{\Pi }{\left(\gamma \right)}_{t}^{t}$ is the identity operator. The composition law$$\hat{\Pi }{\left(\gamma \right)}_{s}^{t}\;\hat{\Pi }{\left(\gamma \right)}_{r}^{s}=\hat{\Pi }{\left(\gamma \right)}_{r}^{t}$$
holds for any smooth non-self-intersecting path in M.

The covariant derivative $\nabla_{\dot{\gamma }}$ of a vector field, *X*, along the path γ in M is given in terms of parallel transport operators by(14)$${\left[\nabla_{\dot{\gamma }}X\right]}_{s}=\frac{\;d\;}{\;dt}\Pi {\left(\gamma \right)}_{t}^{s}\;X_{t}){|}_{t=s}.$$

A similar expression holds for the covariant derivative ${\hat{\nabla }}_{\dot{\gamma }}$, given the complexified parallel transport operators $\hat{\Pi }{\left(\gamma \right)}_{s}^{t}$.

The parallel transport operators ${\hat{\Pi }}^{★}{\left(\gamma \right)}_{s}^{t}$ of the dual connection satisfy, by definition,(15)$${\left(\hat{\Pi }{\left(\gamma \right)}_{s}^{t}\;x,{\hat{\Pi }}^{★}{\left(\gamma \right)}_{s}^{t}\;y\right)}_{t}={\left(x,y\right)}_{s}\;,\;x,y\in H_{s}.$$

Note the use of the complex inner product from Section 4 and *not* the inner product (Equation 9) of Kubo–Mori theory. The corresponding covariant derivative is denoted as ${\hat{\nabla }}_{\dot{\gamma }}^{★}$. It is linked to the covariant derivative ${\hat{\nabla }}_{\dot{\gamma }}$ by(16)$${\left({\hat{\nabla }}_{\dot{\gamma }}x,y\right)}_{t}+(x,{\hat{\nabla }}_{\dot{\gamma }}^{★}y{)}_{t}=\frac{\;d\;}{\;dt}{\left(x,y\right)}_{t}.$$

## 10. Basis Vectors

Choose a basis, ${\left(e_{p}\right)}_{p}$, of the vector fields. The metric tensor *g* is given by(17)$$g_{pq}\left(\theta \right)={\left(e_{p},e_{q}\right)}_{\theta }.$$

Require that each basis vector, $e_{p}\left(\theta \right)$, belongs either to $K_{\theta }$ or $L_{\theta }$. Introduce the basis vectors $f_{p}\left(\theta \right)$ in the complex Hilbert space $H_{\theta }$, defined by$$\begin{array}{cc} f_{p}\left(\theta \right) & =\left(\begin{array}{c} e_{p}\left(\theta \right)\\ 0 \end{array}\right)\;\mathrm{if}\;e_{p}\left(\theta \right)\in K_{\theta },\\ f_{p}\left(\theta \right) & =\left(\begin{array}{c} 0\\ Je_{p}\left(\theta \right) \end{array}\right)=-i\left(\begin{array}{c} e_{p}\left(\theta \right)\\ 0 \end{array}\right)\;\mathrm{if}\;e_{p}\left(\theta \right)\in L_{\theta }. \end{array}$$

These vectors, $f_{p}$, all belong to $K^{⊕}$. They satisfy$$\begin{array}{cc} {\left(f_{p},f_{q}\right)}_{\theta } & =g_{pq}\left(\theta \right)\;\mathrm{if}\;e_{p}\;\mathrm{and}\;e_{q}\in K_{\theta }\;\mathrm{or}\;e_{p}\;\mathrm{and}\;e_{q}\in L_{\theta },\\ \; & =ig_{pq}\left(\theta \right)\;\mathrm{if}\;e_{p}\in K_{\theta }\;\mathrm{and}\;e_{q}\in L_{\theta }. \end{array}$$

From $\hat{S}f_{p}=f_{p}$ one obtains ${\hat{\Delta }}^{1/2}f_{p}=\hat{m}f_{p}$. A short calculation then gives$$\begin{array}{cc} {\left({\hat{\Delta }}^{1/2}f_{p},f_{q}\right)}_{\theta } & =G_{pq}\left(\theta \right)\;\mathrm{if}\;e_{p},e_{q}\in K,\\ \; & =-G_{pq}\left(\theta \right)\;\mathrm{if}\;e_{p},e_{q}\in L,\\ \; & =0\;\mathrm{otherwise}, \end{array}$$
with $G_{pq}$ being the pseudometric introduced in Section 2. This result implies that(18)$${\hat{\Delta }}^{1/2}f_{p}=R_{p}^{q}f_{q}$$
with$$\begin{array}{cc} R_{p}^{q} & ={\left({\hat{\Delta }}^{1/2}f_{p},f_{r}\right)}_{\theta }\;g^{rq}\;\mathrm{if}\;e_{p}\;\mathrm{and}\;e_{q}\in K_{\theta }\;or\;e_{p}\;\mathrm{and}\;e_{q}\in L_{\theta },\\ \; & =i{\left({\hat{\Delta }}^{1/2}f_{p},f_{r}\right)}_{\theta }\;g^{rq}\;\mathrm{if}\;e_{p}\in K_{\theta }\;\mathrm{and}\;e_{q}\in L_{\theta }. \end{array}$$

## 11. Connection Coefficients

Up to this point, no differentiability in the manifold M has been assumed. Now, the assumption is made that the covariant derivatives of the vector fields $e_{p}$ do exist.

Let $\nabla_{p}$ denote the covariant derivative $\nabla_{\dot{\gamma }}$ in the direction of $e_{p}$, i.e., $\dot{\gamma }=e_{p}$. Require that for each smooth path, γ, in the manifold M, the covariant derivatives $\nabla_{\dot{\gamma }}e_{q}$ exist and are given by(19)$$\nabla_{\dot{\gamma }}e_{q}={\dot{\gamma }}^{p}\;\nabla_{p}e_{q}.$$

These covariant derivatives can be expanded in the basis vectors. This gives(20)$$\nabla_{p}e_{q}=\Gamma_{pq}^{r}\;e_{r}\;\mathrm{and}\;\nabla_{\dot{\gamma }}e_{q}={\dot{\gamma }}^{p}\;\Gamma_{pq}^{r}\;e_{r},$$
with the connection coefficients being $\Gamma_{pq}^{r}$.

The r.h.s. of the above equations can be split up into a vector belonging to K and a vector belonging to L. Let$$\Gamma_{pq}^{r}=\Xi_{pq}^{r}+\Theta_{pq}^{r}$$
with $\Xi_{pq}^{r}=0$ when $e_{r}\in L$ and $\Theta_{pq}^{r}=0$ when $e_{r}\in K$. Then, a short calculation, using the definition (Equation 14) of the covariant derivative and using (Equation 20), shows that if $e_{q}\in K$, then(21)$$\begin{array}{cc} {\left[{\hat{\nabla }}_{\dot{\gamma }}f_{q}\right]}_{s} & =\frac{\;d\;}{\;dt}\hat{\Pi }{\left(\gamma \right)}_{t}^{s}\;f_{q}\left(\gamma_{t}\right){|}_{t=s}\\ \; & =\frac{\;d\;}{\;dt}\Pi {\left(\gamma \right)}_{t}^{s}e_{q}\left(\gamma_{t}\right){|}_{t=s}\\ \; & ={\left[\nabla_{\dot{\gamma }}e_{q}\right]}_{s}\\ \; & ={\dot{\gamma }}^{p}\;\left(\Xi_{pq}^{r}+i\Theta_{pq}^{r}\right)\;f_{r}\left(\gamma_{s}\right). \end{array}$$

If $e_{q}\in L$, then one has(22)$$\begin{array}{cc} {\left[{\hat{\nabla }}_{\dot{\gamma }}f_{q}\right]}_{s} & =\frac{\;d\;}{\;dt}\hat{\Pi }{\left(\gamma \right)}_{t}^{s}\;f_{q}\left(\gamma_{t}\right){|}_{t=s}\\ \; & =-i\frac{\;d\;}{\;dt}\Pi {\left(\gamma \right)}_{t}^{s}e_{q}\left(\gamma_{t}\right){|}_{t=s}\\ \; & =-i{\left[\nabla_{\dot{\gamma }}e_{q}\right]}_{s}\\ \; & =-i{\dot{\gamma }}^{p}\;\left(\Xi_{pq}^{r}+i\Theta_{pq}^{r}\right)\;f_{r}\left(\gamma_{s}\right). \end{array}$$

These expressions show in an explicit manner that the connection coefficients of the complexified connection are complex numbers.

## 12. A Fluctuation–Dissipation Theorem

Let it be given that there is a smooth path, γ, in the manifold M. Choose the perturbation operator $\hat{B}$ from Section 8 equal to(23)$${\hat{B}}_{t}=\frac{1}{t-s}\;\left(\hat{\Pi }{\left(\gamma \right)}_{s}^{t}\;{\hat{H}}_{s}\;\hat{\Pi }{\left(\gamma \right)}_{t}^{s}-{\hat{H}}_{s}\right)$$
where $\gamma_{s}$ is a fixed point along the path γ. Let x denote a field of the eigenvectors of the modular operator $\hat{\Delta }$ and assume that the corresponding eigenvalue λ is differentiable along the path γ. Then, (Equation 12) yields, for an arbitrary vector field, y, in $K^{⊕}$,(24)$$\begin{array}{cc} {\left(\frac{\;d\;}{\;dt}\exp\left(\hat{\Pi }{\left(\gamma \right)}_{t}^{s}{\hat{H}}_{t}\hat{\Pi }{\left(\gamma \right)}_{s}^{t}\right){|}_{t=s}x,y\right)}_{s} & ={\left(\lim_{t\to s}\exp\left({\hat{H}}_{s}+\left(t-s\right){\hat{B}}_{t}\right)\;x,y\right)}_{s}\\ \; & =\lambda \chi_{uy}\left(-\log\lambda \right)+i\lambda \chi_{vy}\left(-\log\lambda \right) \end{array}$$
with u and v in $K^{⊕}$ such that $u+iv=\hat{B}x$. Note that the expressions for u and v are given by Proposition 1.

Next, calculate$$\begin{array}{cc} {\left[\nabla_{\dot{\gamma }}\hat{\Delta }x\right]}_{s} & =\frac{\;d\;}{\;dt}\hat{\Pi }{\left(\gamma \right)}_{t}^{s}\;{\hat{\Delta }}_{t}x_{t}{|}_{t=s}\\ \; & =\frac{\;d\;}{\;dt}\exp\left(\hat{\Pi }{\left(\gamma \right)}_{t}^{s}H_{t}\hat{\Pi }{\left(\gamma \right)}_{s}^{t}\right)\;\hat{\Pi }{\left(\gamma \right)}_{t}^{s}x_{t}{|}_{t=s}\\ \; & =\left(\frac{\;d\;}{\;dt}\exp\left(\hat{\Pi }{\left(\gamma \right)}_{t}^{s}H_{t}\hat{\Pi }{\left(\gamma \right)}_{s}^{t}\right)\right){|}_{t=s}\;x_{s}+{\hat{\Delta }}_{s}\;\frac{\;d\;}{\;dt}\hat{\Pi }{\left(\gamma \right)}_{t}^{s}x_{t}{|}_{t=s}. \end{array}$$

Now, use (Equation 24) and$${\left[\nabla_{\dot{\gamma }}\hat{\Delta }x\right]}_{s}=\dot{\lambda }x_{s}+\lambda {\left[\nabla_{\dot{\gamma }}x\right]}_{s}$$
to obtain(25)$$\begin{array}{cc} \dot{\lambda }{\left(x,y\right)}_{s} & ={\left(\hat{\Delta }\;\nabla_{\dot{\gamma }}x,y\right)}_{s}-\lambda {\left(\nabla_{\dot{\gamma }}x,y\right)}_{s}+\lambda \chi_{uy}\left(-\log\lambda \right)+i\lambda \chi_{vy}\left(-\log\lambda \right). \end{array}$$

The l.h.s. of this expression vanishes when the eigenvalue λ is constant along the path γ. If this is the case for all eigenvalues, then the path is said to be *adiabatic*. The first two terms in the r.h.s. represent the change in the eigenvector x along the path. The remaining two terms represent the effect of perturbing the generator *H* of the modular automorphism group.

## 13. Discussion

In the present work, the tangent spaces $T_{\theta }M$ of the Riemannian manifold M are embedded into complex Hilbert spaces, $H_{\theta }$. This is performed in such a way that the two subspaces, $K_{\theta }$ and $L_{\theta }$, of $T_{\theta }M$ become invariant for a modular automorphism group after complexification. The two subspaces correspond with two types of tangent vectors. Any tangent vector is a linear combination of two vectors of different types, and the intersection of the two subspaces is trivial.

In Statistical Physics, the modular automorphism group is important because it describes the time evolution of quantum systems in thermodynamic equilibrium. In this context, the two subspaces correspond with Hermitian and anti-Hermitian operators, respectively [1]. The occurrence of the modular automorphism group in the present, more general context allows us to adapt elements of Statistical Physics. In particular, the admittance function [8], which plays an important role in the Kubo–Mori theory of linear response, is introduced here. It is used to define an inner product (Equation 9), which is the equivalent of the inner product used in Kubo–Mori theory. The admittance function shows up in the result (Equation 25), which can qualify as a fluctuation–dissipation theorem [17,18].

A geometric approach to the modular operator theory was initiated by Rieffel and van Daele [1]. The strength of this approach is that a few basic assumptions about the two types of the tangent vectors of a Riemannian manifold suffice to reach highly non-trivial conclusions.

Many aspects of the geometric approach have not been touched upon in the present paper. An obvious question to be studied is how geodesics behave in the presence of the two types of tangent vectors. What is the relation between geodesics and paths in the manifold which conserve the typing structure of the tangent spaces? Other questions concern dual connections and merit attention because of their importance for statistical models belonging to an exponential family.

## Data Availability

No new data were created or analyzed in this study.

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
