# Peer review of "A Complex Structure for Two-Typed Tangent Spaces"

_entropy, 2025, doi:10.3390/e27020117_

Round 1

Reviewer 1 Report

Comments and Suggestions for Authors

A first difficulty is when dealing with infinite-dimensional manifolds, which involves technicalities that need to be dealt with appropriately. The idea is to use as a line of work throughout the paper the existence, in a Riemannian manifold, of a decomposition of the tangent spaces (and also tangent bundles?), which allows for the definition of projectors on each subspace. So, the author states that there exists a unique orthogonal and Hermitian operator satisfying some properties. Here, one should take some care, since the definition of an orthogonal subspace to a given one, is a delicate matter in infinite dimension. In any case, the tangent space $ T_theta M$ is embedded in a Hilbert space, and after complexification the two subspaces become invariant for a modular automorphisms group. Within this context, the admittance function, which plays an important role in Kubo-Mori theory of linear response, can be introduced and used to define an inner product, which is the equivalent of the inner product used in Kubo-Mori theory. 

The paper contains new methods that may be interesting for future research, but should be reviewed in the light of the following comments:

Some sentences are confusing, for instance “Following [17] the complex number $i$ is defined on $T_\theta M \times T_theta M$ by …”.

The use of infinite dimension manifolds should be carefully revised, since many results that are valid for finite dimension do not work in the infinite context.

The theory of connections should be explored, since the decomposition of the tangent bundle in two vector subbundles could be related with the existence of linear connections on the manifold: what are the relations between these subspaces and the vertical one?

Author Response

LIST OF CHANGES

- Added sentences in the first paragraph of the introduction.

- Added 'finite-dimensional' in the first sentence of Section 2.

- Added references:
    P. R. Halmos, Measure Theory (Van Nostrand, New York, 1950)
    https://hal.science/hal-04856250, https://arxiv.org/abs/2501.04010

ANSWER TO REFEREE 1

• Some sentences are confusing, for instance “Following [17] the complex number
 $i$ is defined on $T_\theta M \times T_theta M$ by …”.

    I change 'Following' into 'In'.

• The use of infinite dimension manifolds should be carefully revised,
since many results that are valid for finite dimension do not work in the infinite context.

    The treatment of infinite-dimensonal manifolds is out of scope of the present work.
    The technicity needed to treat infinite-dimensonal Banach manifolds would make the
    paper less readable. I have added 'finite-dimensional' in the first sentence of Section 2.

• The theory of connections should be explored, since the decomposition of the tangent bundle
in two vector subbundles could be related with the existence of linear connections on the manifold:
what are the relations between these subspaces and the vertical one?

    Generically, the geodesics mix the two types, as can be seen on page 8 from the decomposition of
    the connection coefficients into two non-trivial contributions.
    The connection coefficients become complex fields and require further investigation.

Reviewer 2 Report

Comments and Suggestions for Authors

See attached PDF file

Author Response

LIST OF CHANGES

- Added sentences in the first paragraph of the introduction.

- Added 'finite-dimensional' in the first sentence of Section 2.

- Added references:
    P. R. Halmos, Measure Theory (Van Nostrand, New York, 1950)
    https://hal.science/hal-04856250, https://arxiv.org/abs/2501.04010

ANSWER TO REFEREE 2

• P1-L16 Is it possible to explain briefly the motivation of the term “modular”? 
Or refer to a standard textbook?

  'Modular' in this context comes from the theory of Haar measures.
  I added some sentences in the introduction and a reference to the famous book of Halmos.

• P1-L17 I am unable to access the HAL paper [17]. Is it the same as the
ArXiv https://arxiv.org/abs/2501.04010?

  It is the same. The preprint appeared on arXiv a little bit later.
  The technical problem with accessing the HAL preprint has been resolved.

• P2-FN1 Which part of the theory holds in this infinite dimensional case?

  The theory of the modular operator holds in the infinite case almost without limitations,
  also in the work of Rieffel and van Daele. 
  A few statements in my paper with Jun Zhang 
  need further investigation in the infinite-dimensional case. 
  The present application to for instance infinite-dimensional Banach or Hilbert manifolds 
  requires much more work and has still to be done.

• P3-(3) At this point, to see that i^2 = −1 the reader should know J^2 = I.
Is it because J is orthogonal and Hermitian?

  This follows indeed from J*J=I and J*=J.

• P6-L163 In Amari’s Information Geometry, in some cases, the explicit
expression of the mixture and exponential parallel transports are available
in closed form. In the affine case, the operator does not depend on the
curve γ. Are there cases of explicit expression in Amari’s 0-th case, e.g.,
the Riemannian case? Or would this notion be, in most cases, of theoretical
interest only?

  The Physics example of the Aharony-Bohm effect and of Berry's phase suggests that
  at least in the case of the quantum exponential family path dependence
  does occur. See the paper of Kolodrubetz et al.